# Clinical Characteristics of Adenovirus Pneumonia in Children

**DOI:** 10.3390/pathogens14111110

**Published:** 2025-10-31

**Authors:** Huifen Xu, Wei Chen, Qinrui Lai, Yingying Chen, Yajun Guo, Jing Chen, Wei Li

**Affiliations:** 1Department of Pharmacy, The Children’s Hospital, Zhejiang University School of Medicine, National Clinical Research Center for Child Health, Hangzhou 310053, China; 6515119@zju.edu.cn; 2Department of Clinical Laboratory, The Children’s Hospital, Zhejiang University School of Medicine, National Clinical Research Center for Child Health, Hangzhou 310053, China; 22418786@zju.edu.cn (W.C.); 22318126@zju.edu.cn (Q.L.); 6513085@zju.edu.cn (Y.C.); 3Department of Ultrasound, The Affiliated Huaian No. 1 People’s Hospital of Nanjing Medical University, Huaian 223001, China

**Keywords:** adenoviral upper respiratory infection, children, severe pneumonia, early prediction, random forest

## Abstract

Identifying effective indicators and developing predictive models for the early detection of severe adenoviral pneumonia (SAP) is critical to safeguarding patients’ lives. This study examined differences between 428 patients with SAP and those with non-severe adenoviral pneumonia (NSAP) from March 2022 to January 2023, focusing on variables such as age, sex, type of coinfection, and a range of clinical laboratory indicators. SAP was significantly more common in children aged 3–6 years (20/54 of all SAP cases, *p* = 0.0258) and among those with polymicrobial coinfections (*p* < 1.20 × 10^−11^). Patients with SAP exhibited significantly higher prealbumin (PA) level, while C-reactive protein (CRP) level was significantly lower. Composite indicator, such as CRP -to- prealbumin ratio (CPAR), was also significantly elevated (*p* < 0.05). The random forest model achieved an area under the receiver operating characteristic (ROC) curve (AUC) of 0.699, with an accuracy of 84.5% and a precision of 91.5%. Analysis of the data revealed key predictive parameters for early-stage SAP. Indicators such as CPAR, PA, and CRP are valuable for assessing SAP risk. Moreover, commonly available clinical indicators can effectively construct a random forest-based predictive model for SAP.

## 1. Introduction

Respiratory adenoviral infections present with a wide range of clinical manifestations, from mild upper respiratory symptoms to severe pneumonia, and are a leading cause of hospitalization in children [1,2]. While recent studies have enhanced our understanding of the clinical distinctions between severe adenoviral pneumonia (SAP) and non-severe adenoviral pneumonia (NSAP) in adults, further investigation is needed to elucidate the relationships among clinical characteristics and laboratory indicators in pediatric populations [3,4].

In brief, the pathogenesis of SAP involves a dual mechanism of direct viral injury and excessive host immune response. The virus replicates extensively within airway epithelial cells, directly causing cellular destruction. This process hyperactivates the innate immune system, precipitating a “cytokine storm” that drives the development of diffuse alveolar damage and acute respiratory distress syndrome (ARDS). Consequently, the immune response, while aimed at viral clearance, inflicts significant immunopathological damage on the lung tissue [5].

There is a lack of effective biomarker combinations capable of predicting SAP from NSAP in the early stages of illness. Variables such as age, sex, and types of co-infection may significantly influence disease severity. Furthermore, hematological and biochemical markers—including complete blood count (CBC), prealbumin (PA), and procalcitonin (PCT)—have been emphasized for their roles in assessing infection severity and predicting clinical outcomes [6]. These laboratory parameters are critical for evaluating overall health status, diagnosing diseases, monitoring progress, and guiding therapeutic decisions [7,8,9]. Recent findings have also drawn attention to the significance of inflammatory markers, such as interleukins (IL-2, IL-4, IL-6, IL-10), tumor necrosis factor (TNF), and interferon-gamma (IFN-γ), in the pathogenesis of adenoviral infections [10,11]. These cytokines are instrumental in modulating immune and inflammatory responses, thus providing insights into the patient’s immunological condition and pathological processes. Evaluating these inflammatory biomarkers alongside hematological and biochemical indicators may enhance our understanding of adenovirus-related disease mechanisms in children [12].

Given the complex clinical and biological profile of SAP—and the similar initial symptoms shared by SAP and NSAP, such as high fever and persistent cough—relying solely on individual clinical markers may be insufficient for accurate early-stage diagnosis. Recently, machine learning has been widely applied across various areas of healthcare, including the diagnosis and management of diabetes, cancer, cardiovascular diseases, and mental health conditions [13,14,15]. The use of artificial intelligence in medicine is increasingly being recognized for its potential to enhance clinical decision-making and facilitate personalized treatment approaches [16,17].

This study aims to compare severe and non-severe adenoviral pneumonia in children by analyzing correlations between clinical characteristics and commonly used laboratory indicators. We apply machine learning techniques to evaluate predictive clinical markers and biomarkers associated with SAP, with the goal of developing an early prediction model. Such a model is essential for identifying high-risk patients and may improve clinical understanding and management of SAP.

## 2. Materials and Methods

### 2.1. Study Design and Patients

A total of 428 pediatric patients were included from March 2022 to January 2023, with adenovirus infection confirmed by nasopharyngeal swab testing. Diagnosis was based on the Guidelines for the Diagnosis and Treatment of Community-Acquired Pneumonia in Children (2019 version) (Table A1 in Appendix B) [18]. Inclusion criteria were as follows: (1) age under 18 years; (2) laboratory-confirmed adenovirus infection; and (3) absence of major comorbidities such as malignancies or nephrotic syndrome.

### 2.2. Specimen Testing

Venous blood samples were collected for laboratory testing, all within 24 h of hospital admission. Hematological parameters—including lymphocyte count (LYM), platelet count (PLT), neutrophil absolute count (NEV), and C-reactive protein (CRP)—were analyzed using flow cytometry with a Mindray BC5300 analyzer (Mindray, Shenzhen, China). Th1/Th2 cytokines were measured using a FACScalibur^®^ flow cytometer (Becton Dickinson, San Jose, CA, USA). Biochemical markers, such as prealbumin (PA) and procalcitonin (PCT), were assessed using colorimetric methods on a Beckman Coulter AU5800 analyzer (Beckman Coulter, Brea, CA, USA). All commercially available kits utilized in this study are documented in Table A2 (Appendix C). All tests/assays were performed according to the standard operating procedures of the kits.

### 2.3. Statistical Analysis

We examined the distribution of severe and non-severe adenoviral pneumonia cases across different age groups, sexes, and types of co-infection. The chi-square test was used to determine statistical significance for these categorical variables. For each biomarker, which were continuous variables and did not assume a normal distribution, differences between the SAP and NSAP groups were assessed using the non-parametric Wilcoxon rank-sum test. All tests were two-sided, with statistical significance set at *p* < 0.05. For biomarkers showing significant differences, receiver operating characteristic (ROC) curve analysis was conducted to evaluate diagnostic value, with an area under the curve (AUC) > 0.6 was considered suggestive of potential diagnostic utility. Owing to its high predictive accuracy achieved through bootstrap aggregation and its innate ability to rank predictor contributions, a random forest model was employed to construct the predictive model and assess feature importance based on the mean decrease in Gini impurity. All data analyses and visualizations were performed using R version 4.3.1. The R packages used were: dplyr, tidyr, broom, grid, gridExtra, patchwork, randomForest, caret, pROC, and ggplot2. The statistical code and random forest implementation have been made available in the Appendix A.

## 3. Results

This study included a total of 428 pediatric patients who tested positive for human adenovirus (HAdV). Among them, 374 were diagnosed with non-severe adenovirus pneumonia (NSAP), and 54 were diagnosed with severe adenovirus pneumonia (SAP). Data revealed significant differences between the SAP and NSAP groups in terms of demographic characteristics (e.g., a higher incidence in children aged 3–6 years), laboratory parameters, and coinfection status. Specifically, the SAP group exhibited significantly higher levels of prealbumin (PA), and C-reactive protein-to-prealbumin ratio (CPAR), while the level of C-reactive protein (CRP) was significantly lower. Furthermore, mixed infections were more likely to lead to SAP. Based on these distinguishing factors, we subsequently constructed an early predictive model for SAP using the random forest algorithm, with the detailed results presented in the following sections.

### 3.1. Patient Characteristics

We collected data from 428 HAdV-positive pediatric patients. Based on the criteria outlined in Table A1, patients were categorized according to the severity of their condition. Among them, 374 were classified as having NSAP, and 54 as having SAP. Patients were grouped based on age, sex, and type of co-infection. According to coinfection status, patients were stratified into five categories: HAdV monoinfection; HAdV–bacterial coinfection; HAdV–other viral coinfection; HAdV–atypical pathogen coinfection (Chlamydia/Mycoplasma); and polymicrobial coinfection (≥2 additional pathogens concurrent with HAdV infection). The case numbers and proportions for each group are summarized in Table 1. The age group with the highest SAP rate was 3–6 years (*p* = 0.0258). The polymicrobial coinfection group exhibited the highest severity rate (*p* = 1.20 × 10^−11^) (Table 1 and Figure 1). No statistically significant difference in SAP occurrence was observed based on sex (*p* = 0.1509).

### 3.2. Variables of Importance

We analyzed the following biomarkers in all patients: lymphocyte count (LYM), prealbumin (PA), C-reactive protein (CRP), platelet count (PLT), neutrophil absolute count (NEV), procalcitonin (PCT), interleukins (IL-2, IL-4, IL-6, IL-10), tumor necrosis factor (TNF), and interferon-gamma (IFN-γ). We also evaluated several composite indicators, including PNR (PLT/NEV), NLR (NEV/LYM), PLR (PLT/LYM), and CPAR (CRP/PA). Reference ranges, means, and standard deviations for these indicators are listed in Appendix D. The Wilcoxon rank-sum test revealed no significant differences in LYM (*p* = 0.4158), NEV (*p* = 0.6049), PCT (*p* = 0.0970), IL-2 (*p* = 0.0932), IL-4 (*p* = 0.5260), IL-6 (*p* = 0.0766), IL-10 (*p* = 0.9097), or TNF (*p* = 0.3006). However, significant differences were observed for PA (*p* = 0.0009), CRP (*p* = 0.0005), PLT (*p* = 0.0405), and IFN-γ (*p* = 0.0087). Among the composite indicators, PNR (*p* = 0.5942) and NLR (*p* = 0.2481) showed no significant differences, while PLR (*p* = 0.0184) and CPAR (*p* = 0.0000) were significantly different (Figure 2 and Table A3 in Appendix D).

To assess diagnostic value, ROC curve analysis was conducted on all indicators with statistically significant differences. Receiver operating characteristic (ROC) analysis was employed to quantify the diagnostic value of these significant biomarkers. The area under the ROC curve (AUC) serves as a key metric, where an AUC > 0.6 confirms their utility as potential diagnostic aids, with higher values indicating stronger predictive power for severe disease. As shown in Figure 3, the following AUC values were observed: PA (AUC = 0.641), CRP (AUC = 0.647), PLT (AUC = 0.586), IFN-γ (AUC = 0.67), PLR (AUC = 0.599), and CPAR (AUC = 0.677).

### 3.3. Random Forest Model

On the independent test set, the model demonstrated moderate discriminatory power (AUC = 0.699), with high sensitivity for NSAP cases (91.5%) but limited specificity (16.7%) (Figure 4, Table 2), reflecting the inherent class imbalance in the dataset. Initial univariate analysis using Wilcoxon rank-sum test identified several biomarkers with significant differences between severity groups: PA, CRP, PLT, IFN-γ, PLR, and CPAR. However, to control for multiple testing and reduce false discoveries, we applied random forest FDR correction. After FDR adjustment, only four features remained statistically significant (FDR < 0.05): CPAR (FDR *p* = 0.002), CRP (FDR *p* = 0.01), PA (FDR *p* = 0.009), and IL-10 (FDR *p* = 0.043). Notably, PLT and IFN-γ, which showed significance in the unadjusted analysis, did not survive random forest FDR correction (FDR *p* = 0.082 and 0.103, respectively).

Consequently, feature selection for the final random forest model was based strictly on the FDR-corrected significance threshold, incorporating only the four features that met this criterion. As illustrated in Figure 5, feature importance analysis within the final model identified CPAR, CRP and PA as the most influential predictors, with higher Gini importance values indicating greater contribution to the prediction of disease severity.

## 4. Discussion

The data from this study show that the proportion of severe cases among children infected with adenovirus was approximately 12.3%, notably higher than the reported severe case rate of approximately 5% for COVID-19 and exceeding the 1–10% range typically associated with influenza [19,20,21]. We observed that the highest proportions of adenoviral pneumonia cases were in children aged 3–6 years (45.7%) and 6–9 years (35.0%). We observed a difference in the age distribution in our cohort compared to previous literature, which commonly identifies infants and toddlers (6 months to 3 years) as the most affected group [22]. These results emphasize the need for heightened clinical awareness of adenoviral infections in children aged 3–6 years and increased vigilance in monitoring changes in their clinical condition. We interpret this finding with caution, as the cross-sectional nature of our data does not allow us to rule out the influence of local epidemiological factors or chance variation. The discrepancy may be attributed to regional differences in study populations, sampling bias, or variations in sample size. Consistent with prior studies [23], our data showed no significant sex-based differences in infection or severity (*p* = 0.1509).

Previous research has suggested that coinfection with Mycoplasma pneumoniae and elevated viral loads may serve as risk factors for severe adenoviral pneumonia in immunocompetent children [24]. In our study, HAdV monoinfection accounted for the highest proportion of positive cases (56.54%), while the polymicrobial coinfection group—defined as the presence of three or more additional pathogens—exhibited the highest severity rate, with 40.7% of cases classified as severe (*p* = 1.20 × 10^−11^), a statistically significant finding. This underscores the importance of closely monitoring cases involving multiple pathogens in clinical settings. Accordingly, clinicians should maintain a high index of suspicion for polymicrobial coinfections in patients with HAdV infection, ensure careful tracking of disease progression, and consider prompt administration of broad-spectrum antimicrobial therapy when clinically indicated.

We found statistically significant differences in several key biomarkers between SAP and NSAP patients. Specifically, PA and PLT levels were significantly higher in SAP patients, while CRP and IFN-γ levels were significantly lower. Three indicators—CRP (AUC = 0.647), PA (AUC = 0.641), and IFN-γ (AUC = 0.670)—achieved AUCs above 0.6, indicating potential predictive value, whereas PLT performed less well (AUC = 0.586). Elevated PLT may indicate vascular involvement in severe cases [25,26,27], although measured CRP levels in both NSAP and SAP were higher than those in healthy children, suggesting a complex inflammatory response [28,29,30]. Alterations in PA likely reflect changes in hepatic function or nutritional status, and reduced IFN-γ may be indicative of immune dysregulation [31,32]. Moreover, we found that while IFN-γ levels differed significantly between NSAP and SAP patients, most values fell within the reagent’s reference interval. This finding highlights the need to establish pediatric-specific reference intervals for IFN-γ and warrants further investigation into its role in the pathogenesis of adenovirus pneumonia. Monitoring these biomarkers—CRP, PA, and IFN-γ—can assist in evaluating the risk of severe disease and tailoring therapeutic strategies accordingly. In addition to individual biomarkers, we explored composite indicators to enhance diagnostic accuracy. Previous studies have shown that combined use of TNF-α and IL-6, or composite platelet-related indices, may improve diagnostic performance [33]. Inspired by this, we conducted a combined analysis and identified significant differences in the platelet-to-lymphocyte ratio (PLR) and CRP-to-prealbumin ratio (CPAR) between SAP and NSAP groups. Furthermore, ROC analysis revealed AUCs of 0.677 for CPAR and 0.599 for PLR, suggesting that CPAR may possess greater predictive value. This composite marker, when used in conjunction, may support early identification of severe cases, inform clinical decision-making, and contribute to more personalized care strategies. However, further validation in larger, multi-center cohorts is needed to confirm their clinical utility and reliability.

The early identification of SAP remains clinically challenging, creating opportunities for machine learning tools to support clinical decision-making [34,35]. In this study, we developed a random forest model to predict SAP occurrence, implementing several methodological refinements to enhance robustness: FDR correction for multiple testing, up-sampling to address class imbalance, and repeated cross-validation. The model achieved an AUC of 0.699 on the independent test set, with performance metrics detailed in Table 2. Although its sensitivity is as high as 0.915, the combined performance metrics—including an Accuracy of 0.845 and a Specificity of 0.167—lead us to recognize that, due to imbalanced datasets, the overall performance remains at a moderate level. Therefore, the model currently only holds potential utility as a screening tool. Feature importance analysis identified CPAR as the strongest predictor, followed by PA and CRP. These findings, while derived from a single-center dataset, provide a methodological foundation for future multi-center studies with larger sample sizes.

We observed that the predictors highlighted by the random forest model differed from those with significant univariate differences. This is an expected phenomenon, as univariate tests identify distinct features, while multivariate models select for features that provide unique, non-redundant predictive power. Features that showed significance in both analyses are considered robust and important markers, as they not only exhibit distributional differences between groups but also hold significant predictive power in the multivariable model. The prominence of the composite index CPAR over individual markers like CRP and PA in our model underscores the value of integrative biomarkers that capture multiple physiological pathways, potentially offering a more robust basis for clinical prediction tools.

Our study integrated data from clinical history, physical examinations, and laboratory tests to identify statistically significant predictors of severe adenoviral pneumonia. Based on these findings, we constructed a machine learning model to predict SAP, offering an innovative approach by quantifying the predictive value of laboratory biomarkers. A larger pool of candidate variables and a greater number of analyzed cases were positively correlated with improved model accuracy. Given that adenoviral pneumonia is one of the most severe types of pediatric pneumonia [36], our approach holds practical clinical relevance. While the model’s high sensitivity suggests it may contribute to early identification by helping to flag children at higher risk, its clinical utility for directly improving outcomes requires further validation in external populations.

However, this study has several limitations that should be considered when interpreting the results. First, as a single-center study, our findings may be influenced by institutional biases and patient selection practices, which could affect the generalizability of our model. This study is also limited by its retrospective design. Our case selection criterion, which required the first positive HAdV test within 24 h at our center, crucially cannot account for early interventions or treatments that patients might have received prior to referral. Moving forward, model refinement would benefit from a prospective approach that enrolls incident cases at presentation, with careful documentation of their illness history, rather than relying on historical data audits. Second, the lack of adenovirus genotyping data prevents us from analyzing how specific serotypes, such as the highly pathogenic AdV-B7 [37], may impact disease severity and model predictions. Furthermore, the absence of serum viral load measurements, a recognized predictor of pneumonia severity [38,39], limits our ability to fully assess the determinants of severe disease and may represent an unaccounted-for variable in our predictive algorithm. Additionally, for certain biomarkers, we relied on reagent reference intervals due to a lack of measured values from a healthy pediatric control group within the same region; future studies incorporating such controls would provide more compelling evidence.

These findings underscore the importance of including adenovirus genotyping and viral load quantification in routine diagnostics. Special attention should be given to highly pathogenic strains like AdV-B7 during treatment planning. Our future goals include expanding the number of predictive indicators and case samples from multiple centers to refine the model and improve its predictive performance for severe adenoviral pneumonia.

## 5. Conclusions

In this study, severe adenoviral pneumonia primarily affected children aged 3–6 years. Compared with non-severe cases, patients with SAP showed significantly higher levels of PA and CPAR, while the level of CRP was notably lower. Furthermore, children with mixed infections were more likely to progress to severe disease following adenovirus infection. These indicators, when combined in a random forest model, form a predictive tool that may assist in early SAP identification and support clinical risk management.

## Figures and Tables

**Figure 1 pathogens-14-01110-f001:**
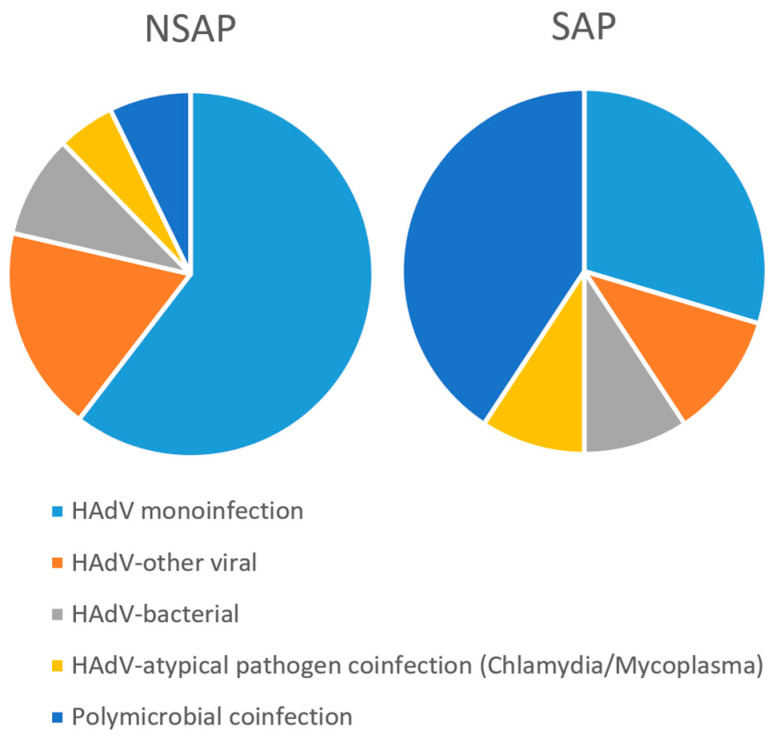
Pie Chart of Infection Types.

**Figure 2 pathogens-14-01110-f002:**
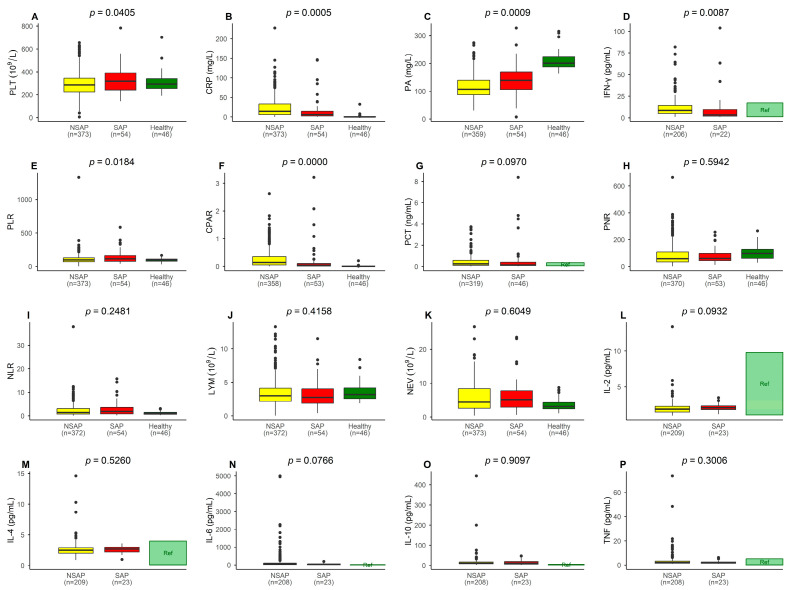
Comparison of laboratory indicators between the SAP and NSAP groups. The parameter measured in each panel is indicated on its left side. Data from healthy children are included for reference. For indicators where measured values from healthy children were unavailable, the reference intervals of the detection reagents are presented and labeled as “Ref” in the graph. Panels (**A**–**F**): statistically significant differences (*p* < 0.05); Panels (**G**–**P**): no statistically significant differences (*p* > 0.05).

**Figure 3 pathogens-14-01110-f003:**
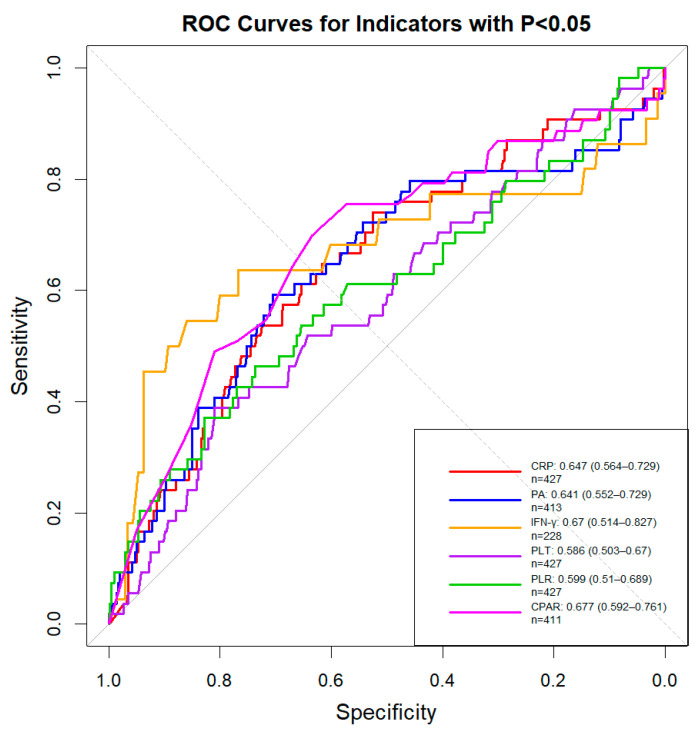
ROC curves for PLT, CRP, PA, IFN-γ, PLR, and CPAR. The AUC value for each indicator is labeled on the graph with its corresponding color-coded number. The values in parentheses in the legend represent the 95% confidence intervals (CI) for each indicator. The number of cases (n) for each indicator was subsequently labeled.

**Figure 4 pathogens-14-01110-f004:**
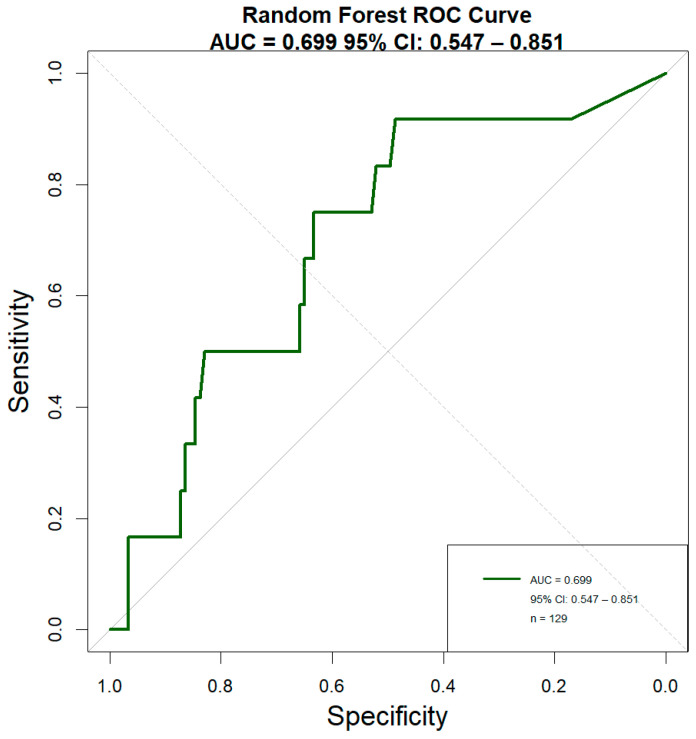
ROC analysis of random Forest models for early identification of SAP. The ROC curves illustrate the performance of the random forest classifier in distinguishing between non-severe/severe adenoviral pneumonia (NSAP/SAP). An identical AUC of 0.699 demonstrates the robust and consistent discriminatory power of the models.

**Figure 5 pathogens-14-01110-f005:**
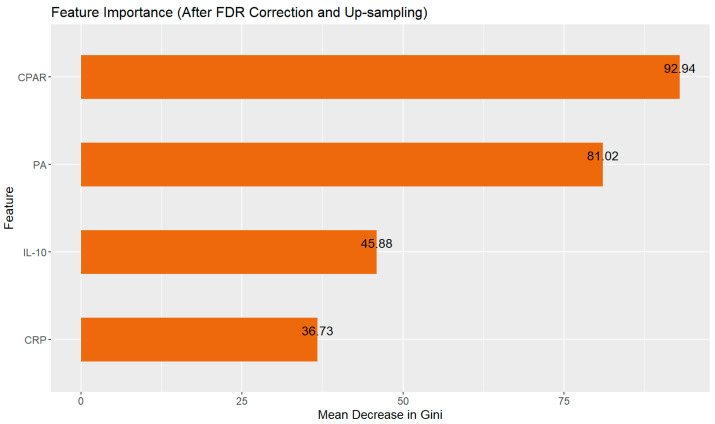
Feature importance analysis of biomarkers for SAP prediction. Bars represent the mean decrease in Gini index, indicating each feature’s contribution to the random forest classification model. CPAR shows the highest predictive importance (82.9), substantially outperforming PA (45.88) and CRP (36.73) following FDR multiple testing correction.

**Table 1 pathogens-14-01110-t001:** Comparison of clinical characteristics between SAP and NSAP in 428 children.

Parameter	Classification	NSAP (*n* = 374) ^1^	SAP (*n* = 54) ^1^	χ	*p*
Gender	Male	221 (59.1)	38 (70.4)	2.0626	0.1509
	Female	153 (40.9)	16 (29.6)		
Age ^2^	>6 M and ≤3 Y	33 (8.8)	12 (22.2)	9.2758	0.0258 *
	>3 Y and ≤6 Y	171 (45.7)	20 (37.0)		
	>6 Y and ≤9 Y	131 (35.0)	16 (29.6)		
	>9 Y	39 (10.4)	6 (11.1)		
Polymicrobial coinfection ^3^	HAdV monoinfection	226 (64.4)	16 (28.1)	57.0658	1.20 × 10^−11^
	HAdV-other viral	68 (18.2)	6 (11.1)		
	HAdV-bacterial	34 (9.1)	5 (9.3)		
	HAdV-atypical pathogen coinfection (Chlamydia/Mycoplasma)	19 (5.1)	5 (9.3)		
	Polymicrobial coinfection	27 (7.2)	22 (40.7)		

Note: ^1^ Data are presented as *n* (%). ^2^ Age group classification is based on the article “Clinical Analysis of Infectious Mononucleosis in Different Age Stages of Children.” ^3^ Definitions of infection types: HAdV monoinfection: adenovirus infection only; HAdV-bacterial coinfection: adenovirus with bacterial coinfection; HAdV-other viral coinfection: adenovirus with other viral pathogens; HAdV-atypical pathogen coinfection: adenovirus with Mycoplasma or Chlamydia coinfection; Polymicrobial coinfection: adenovirus with two or more additional pathogens of different types (e.g., bacterial, fungal, or other viral). *: A *p* value < 0.05 is denoted by an asterisk (*) and considered statistically significant.

**Table 2 pathogens-14-01110-t002:** Comparative performance metrics of the random forest classification model.

Metric	Value
Accuracy	0.845
Precision	0.915
Sensitivity	0.915
F1-Score	0.915
Specificity	0.167
AUC	0.699
Balanced Accuracy	0.541
MCC	0.081

## Data Availability

The datasets analyzed during the current study are available from the corresponding author upon reasonable request.

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
