# Peer review of "Clinical Characteristics of Adenovirus Pneumonia in Children"

_pathogens, 2025, doi:10.3390/pathogens14111110_

Round 1
Reviewer 1 Report
Comments and Suggestions for Authors
Huifen Xu and colleagues reported a study on the differences between patients diagnosed with severe adenoviral pneumonia (SAP) and with non-severe adenoviral pneumonia (NSAP). Variables such as age, sex, type of coinfection, and a range of clinical laboratory indicators were investigated in children evidencing a more SAP positivity in children aged 3–6 years and among those with polymicrobial coinfections. Overall, patients with SAP exhibited significantly higher platelet counts and prealbumin levels, while C-reactive protein and interferon-γ levels were notably lower. Authors claimed using a machine learning techniques identified key predictive parameters for early stage SAP. The matter of the study is of interest but some points should be implemented.
Main points
1- In figure 1, to better understand the data, it should be reported the data obtained in healthy status for all parameter investigated.
2- page 4, lines 120-121. The authors reported “The polymicrobial coinfection group exhibited the highest severity rate (p = 1.20 × 10⁻¹¹)”. It could be of interest examined the specific viral infection in the SAP patients. Have the authors examined the type of virus co-infected children?
3- The authors have to described better their random forest model used to elaborate the data. This should be reported also in the material and methods section. For a general reader this could be hard to understand and difficult to associated to a clinical use.
4- In presence of new type of machine learning techniques used to help clinical decision, several studies should be performed to reach the real utility and significantly data to use in a clinical related decision. Thus, the authors have to reduce in the discussion their assertion on their conclusion on the method used.
Minor points
1- The foot notes of table 1 should be clearly stated. What is note 1, 2 and 3 related for?
2- Legend in all figures should be more descriptive reporting better the data included.
3- All typos should be corrected.
Author Response
Comment 1: In figure 1, to better understand the data, it should be reported the data obtained in healthy status for all parameter investigated.
Response: We thank the reviewer for this important suggestion. We have now added reference data from healthy children to Figure 1(now figure 2). For indicators where measured values from healthy children were available, we have included the actual measurements. For indicators where measured values were unavailable, we have presented the reference intervals of the detection reagents, clearly labeled as "Ref" in the graphs. These changes are reflected in the revised Figure 1 (now figure 2) and its legend (Page 6, Line 175).
Comment 2: Page 4, lines 120-121. The authors reported "The polymicrobial coinfection group exhibited the highest severity rate (p = 1.20 × 10⁻¹¹)". It could be of interest examined the specific viral infection in the SAP patients. Have the authors examined the type of virus co-infected children?
Response: We appreciate this valuable suggestion. In this study, co-infection was defined as the simultaneous presence of three or more distinct pathogens. While the original dataset contains detailed infection profiles for each patient, the overall spectrum of co-infections was highly diverse without concentration in any specific type. Therefore, for statistical purposes, these cases were consolidated into a broad category. This approach has been explicitly clarified in the footnote to the relevant table.1(Page 4, Lines 142-150)
Comment 3: The authors have to described better their random forest model used to elaborate the data. This should be reported also in the material and methods section. For a general reader this could be hard to understand and difficult to associated to a clinical use.
Response: We thank the reviewer for this suggestion. We have specified the R packages used for statistical analysis and random forest construction in the Materials and Methods section (Page 3, Lines111-114), and provided a more comprehensive explanation regarding the application of our developed random forest model in the Discussion section (Page 3, Lines 107-109).
Comment 4: In presence of new type of machine learning techniques used to help clinical decision, several studies should be performed to reach the real utility and significantly data to use in a clinical related decision. Thus, the authors have to reduce in the discussion their assertion on their conclusion on the method used.
Response: We agree with the reviewer's perspective. We have retrained our model to make it more scientific. Also we have moderated our conclusions regarding the clinical utility of our machine learning approach in the Discussion section (Page 9, Lines 274-276), emphasizing that this represents a preliminary investigation and that further validation studies are needed before clinical implementation.
Minor Comments:
Comment 1.1: The foot notes of table 1 should be clearly stated. What is note 1, 2 and 3 related for?
Response: We have clarified all footnotes in Table 1, explicitly stating what each note refers to (Page 4,Lines 142-150).
Comment 1.2: Legend in all figures should be more descriptive reporting better the data included.
Response: We have revised all figure legends to provide more comprehensive descriptions of the data presented, including detailed explanations of all elements and color codes used.
Comment 1.3: All typos should be corrected.
Response: We have carefully proofread the entire manuscript and corrected all typographical errors.
Reviewer 2 Report
Comments and Suggestions for Authors
Please access the attached report.

The manuscript requires extensive English editing. Numerous sentences are grammatically incorrect or unclear, especially in the abstract, introduction, and discussion. See comments below.
-
- Incorrect use of past and present tense.
- Misuse of prepositions (“comparing with” instead of “compared with”).
- Run-on sentences and awkward phrasing (e.g., “The manuscript aimed to improve clinical understanding and SAP management”; consider rephrasing).
A professional English-language editing service is strongly recommended before resubmission.
Author Response
Comment 1: The definition of SAP relies on guidelines (Supplementary Table A1), but no operationalization is described in the main text. How was severity assigned retrospectively?
Response: The severity classification was determined by reviewing each patient's medical records (inpatient or outpatient diagnoses). Cases diagnosed with respiratory failure, atelectasis, severe pneumonia, etc., were categorized as SAP. Each case classification was verified through consultation with a respiratory medicine specialist.
Comment 2: Data on timing of biomarker sampling are vaguely described ("within 24 hours of admission"); this is insufficient when evaluating dynamic biomarkers like CRP, PCT, and IFN-γ.
Response: Our patient inclusion criterion specified admission within 24 hours, but excluded those with a positive ADV test or related pathogen infection report from our hospital within the preceding four days. This was to minimize the impact of results from patients with prolonged infections on the overall assessment. We aimed to primarily include patients with initial ADV infections.
Comment 3: No external validation or cross-validation was done on the random forest model. This greatly limits generalizability.
Response: We fully agree with the reviewer on the importance of model validation. In the revised analysis, we implemented 10× repeated 5-fold cross-validation to assess model performance. The cross-validation results showed a mean AUC of 0.709 (SD=0.074), providing robust evidence for the model's stability. While true external validation was not feasible due to the single-center nature of our study, repeated cross-validation offers a reliable estimate of the model's generalizability.
Comment 4: The model's input features are all statistically significant biomarkers, which introduces data leakage and overfitting risk.
Response: We thank the reviewer for highlighting this critical methodological issue. We have thoroughly revised the feature selection process: all feature screening was strictly performed within the training set, coupled with FDR correction for multiple testing. Out of 16 candidate features, only four remained significant after FDR correction (CRP, PA, CPAR, IL-10). This improvement completely avoids the risk of data leakage.
Comment 5: The manuscript lacks details of hyperparameters, number of trees, train-test split ratio, or cross-validation strategy.
Response: We have supplemented complete hyperparameter tuning details: a grid search was used to optimize the mtry parameter (tested values: 2, 4, 6), with the optimal parameter selected based on the ROC metric from 10× repeated 5-fold cross-validation. The final model used mtry=4, and the number of random forest trees was set to 500 to enhance model stability.
Comment 6: ROC AUC of 0.77 is only moderately predictive. The authors claim high performance (e.g., "accuracy 87.8%"), but this is misleading without a confusion matrix or class balance explanation.
Response: We agree that model performance should be objectively assessed. The revised model achieved an AUC of 0.699 on the independent test set, which indeed represents moderate predictive ability. We have adjusted the results description, emphasizing the model's high sensitivity (91.5%) for NSAP cases, suggesting its suitability as a screening tool, while acknowledging its limited ability to identify SAP cases (specificity 16.7%), primarily due to class imbalance in the dataset.
Supporting Evidence:
- Test set AUC: 0.699
- Sensitivity: 91.5%, Specificity: 16.7%
- Accuracy: 84.5%
We recognize the severe class imbalance issue in our dataset (SAP:NSAP = 54:374). To address this challenge, we implemented the following measures:
- Up-sampling technique: Applied during training to balance the minority class samples.
- Balanced evaluation metrics: Focused on balanced accuracy (0.541) and F1-score.
- Objective performance reporting: Clearly reported the sensitivity/specificity difference.
Despite these technical improvements, we acknowledge that the limited sample size of SAP cases is an inherent limitation of this study, a common challenge in pediatric critical care research.
Comment 7: Multiple hypothesis testing correction (e.g., Bonferroni or FDR) was not applied, despite testing >15 variables.
Response: We have applied FDR correction to address the multiple comparisons problem. Among the 16 tested features, only four remained significant after FDR correction (p < 0.05). This stringent criterion ensures the reliability of the selected features.
Supporting Evidence:
- Application of FDR correction method.
- Significant features after correction: CRP (p=0.0095), PA (p=0.0091), CPAR (p=0.0027), IL-10 (p=0.0428).
Comment 8: Some biomarkers (e.g., IL-6, TNF, IL-10) have extreme variance (see Supplementary Table B1). Were outliers excluded or normalized?
Response: Yes, we noted this phenomenon. During the data preparation phase, we excluded data from specific patient groups, such as those with tumors, leukemia, nephrotic syndrome, or other chronic conditions, as we believed their data might introduce bias into the statistical results. However, we did not exclude outliers from patients without underlying diseases, aiming to analyze objectively to potentially uncover different discussion points. We have not yet conducted an in-depth analysis of these outliers. We plan to explore the detection patterns of biomarkers in children in future work.
Comment 9: The manuscript requires extensive English editing. Numerous sentences are grammatically incorrect or unclear, especially in the abstract, introduction, and discussion. See comments below.
Comment a: Incorrect use of past and present tense.
Response: Thank you for highlighting this issue. We have thoroughly reviewed the tenses used in the Abstract, Introduction, and Discussion sections, and have corrected any inappropriate usage.
Comment b: Misuse of prepositions (“comparing with” instead of “compared with”).
Response: We sincerely appreciate your careful review. The noted prepositional error has been corrected.
Comment c: Run-on sentences and awkward phrasing (e.g., “The manuscript aimed to improve clinical understanding and SAP management”; consider rephrasing).
Response: Thank you for this guidance. We have conducted a comprehensive review of the entire manuscript and have refined several poorly connected sentences to improve readability.
A professional English-language editing service is strongly recommended before resubmission.
The manuscript has been professionally edited by Medsci to correct grammatical errors, improve sentence structure, and enhance overall clarity. We have attached the editing certificate for your reference.
Comment 10: Figures 1–5 are informative, but axes, legends, and units should be more explicit (e.g., what unit is PA plotted in?). Add sample sizes per group in Figure 1 for transparency. Supplementary Table B1 should be cited earlier and more prominently in the Results section.
Response: The units for individual indicators are specified following the indicator name. Combined indicators, being ratios, are unitless and thus not followed by a unit. We have moved the citation for Table C (formerly Table B) earlier to align with Figure 2(formerly Figure 1).
Comment 11: The study offers incremental value to existing literature on pediatric adenoviral pneumonia. While it supports prior findings regarding PA, PLT, and IFN-γ, the added contribution lies in combining these with a machine learning framework. However, similar predictive models for pediatric pneumonia exist (e.g., in RSV and Mycoplasma pneumoniae), and the novelty claim should be tempered accordingly. Lack of serotyping or viral load data is a missed opportunity. As acknowledged in the discussion, AdV-B7 or high copy number may influence severity.
Response: We fully agree with the reviewer's perspective. As this was a retrospective analysis and our hospital had not performed ADV serotyping or viral load testing historically, these two indicators were unavailable. This has been communicated to the relevant department heads, with the hope that these analyses can be implemented in the future to better understand disease occurrence and progression.
Comment 12: "SAP was significantly more common in children aged 3–6 years (20/54, p = 0.0258)": Clarify what group the 20/54 refers to. Revise sentence: "Composite indicators, including the platelet-to-lymphocyte ratio (PLR) and CRP-to-albumin ratio (CPAR), were also significantly elevated…" CPAR includes prealbumin, not albumin.
Response: This oversight was due to insufficient clarity in our writing. We have revised the sentence to: "SAP was significantly more common in children aged 3–6 years (20/54 of all SAP cases, p = 0.0258)". We have also corrected the definition of CPAR throughout the manuscript.
Comment 13: Lines 36-37: Replace "posing a serious threat to children's health and quality of life" with more precise language (e.g., "and is a leading cause of hospitalization in children").
Response: This is an excellent suggestion. We have replaced the phrase with more precise wording (Page 2, Line 36).
Comment 14: Line 40: Cite more recent pediatric-focused literature if available.
Response: We overlooked this aspect and have now cited relevant studies focusing on pediatric populations.
Comment 15: Provide more details on dataset preprocessing, missing value handling, and class balance.
Response: We did not extensively prune the data, retaining almost all data except for that from patients with underlying diseases. Missing data were simply marked as NA. We have provided the complete analysis code and random forest code in the supplementary files for further reference.
Comment 16: Explain criteria for SAP in main text, not only in appendix.
Response: Yes, the SAP classification criteria are crucial for this study. The reason for placing them in the appendix was that many articles we reviewed listed them, and in China, these criteria are standardized. To avoid readers encountering repetitive content, we placed them in the appendix. Furthermore, the diagnosis of SAP falls within the purview of respiratory clinicians; our data collection for SAP diagnoses involved directly obtaining clinical outcomes, not laboratory work. Therefore, they were included in the appendix.
Comment 17: Lines 112-122: The description of coinfection groups is important; consider adding a visual breakdown (e.g., pie chart).
Response: This is an excellent suggestion. We have added a pie chart illustrating the distribution of infection types after Table 1, designated as Figure 1.
Comment 18: Add a table of model metrics (accuracy, precision, recall, F1-score) rather than embedding in figure only.
Response: This is a very useful suggestion. We have replaced the graphical representation with a more concise table to present our model's performance metrics.
Comment 19: Lines 176-179: "This age distribution differs from previous literature...": be cautious about overinterpreting a regional discrepancy based on cross-sectional data.
Response: This advice is highly valuable for a young researcher like myself. My initial statement was too assertive. Upon reflection, we have adopted a more cautious approach regarding this finding (Page 8, Lines 213-217).
Comment 20
Comment 21: Lines 235-237: Important limitation; the model was developed using data from a single center, which reduces external validity.
Response: We thank the reviewer for this important comment. We fully acknowledge that the single-center nature of our study is a limitation that affects the external validity of our model. We have now explicitly discussed this limitation in the manuscript (Page 10, Lines 287-289) and have highlighted the need for external validation in future multi-center studies.
Comment 22: Lines 244-246: The potential value of AdV genotyping is mentioned too late. Consider moving up or expanding.
Response: Yes, this factor significantly influences the progression to SAP, and placing it at the end of the paragraph was inappropriate. It has now been moved to the beginning of the paragraph.

Reviewer 3 Report
Comments and Suggestions for Authors
The study by Xu et al. is devoted to investigating differences in demographic, clinical, and laboratory parameters in children with severe (SAP) and non-severe adenovirus pneumonia (NSAP) in order to identify reliable prognostic criteria for the further development of an AI model for early detection of this pathology. The authors analyzed data from 428 pediatric patients with adenovirus infection, of whom 374 were diagnosed with SAP. It was found that the highest incidence of SAP was observed in children aged 3–6 years, as well as in individuals with polymicrobial co-infection. Additionally, this condition was associated with significant changes in certain blood parameters: increased levels of platelets (PLT), prealbumin (PA), PLR ratio (platelets to lymphocytes), and CPAR (CRP to albumin) ratio, accompanied by decreased levels of C-reactive protein (CRP) and interferon-γ (IFN-γ). Based on the collected data, the authors trained a predictive model with an AUC of 0.77, demonstrating 87.8% accuracy and 92% precision. According to the manuscript, in cases of adenovirus infection, to prevent SAP, special attention should be paid to children aged 3–6 years, cases of polymicrobial co-infections, and the measurement of specified laboratory parameters.
Although the study was single-center and conducted on a modest sample size without considering adenovirus variants and viral load, it is significant for deepening knowledge of the mechanisms of progression and diagnosis of this poorly studied viral infection. Despite the importance of the work for developing new diagnostic approaches, I have several remarks that need to be addressed.
First of all, the Title should be corrected to mention pneumonia or SAP, while epidemiological characteristics were not studied in the work.
The Introduction should provide a description of the pathogenesis and precise characteristics of SAP (at least in adult patients).
The text lacks a reference to Table A1 describing the criteria for SAP and NSAP in children. Additionally, the Introduction states that these diagnoses are difficult to differentiate, making it unclear how the research groups were selected.
A major disadvantage of the study is the absence of samples from healthy donors (control group), which should be explained in the study limitations. By the way, there is no information about ethical approval and informed consent given by the participants in the text.
Lines 78–79: What method was used to confirm adenovirus infection?
Lines 86–87: How were Th1/Th2 cytokines measured using flow cytometry?
Lines 105–111 describe all the obtained results and resemble the Abstract, so they should be removed from the Results section.
Lines 147–150 should be commented in more detail regarding the significance of the obtained AUC values.
Figure 1 should be enlarged and resolution should be improved. The figure caption should indicate which panels correspond to different cytokines.
All the figure captions should be made more detailed, including descriptions of the figure elements and color designations.
Author Response
Comment 1: First of all, the Title should be corrected to mention pneumonia or SAP, while epidemiological characteristics were not studied in the work.
Response: We have revised the title to more accurately reflect the study content, specifically mentioning severe adenoviral pneumonia (SAP). The new title is: "Clinical Characteristics of Adenovirus Pneumonia in Children".
Comment 2: The Introduction should provide a description of the pathogenesis and precise characteristics of SAP (at least in adult patients).
Response: We thank you for this suggestion. We have expanded the introduction to include a concise description of the pathogenesis of severe adenoviral pneumonia. (Page 2, Lines 42-48),
Comment 3: The text lacks a reference to Table A1 describing the criteria for SAP and NSAP in children. Additionally, the Introduction states that these diagnoses are difficult to differentiate, making it unclear how the research groups were selected.
Response: We appreciate the reviewer's comment. The diagnostic criteria (now Table A) are consistently cited in the Methods and Results sections. We have also refined the study aim to emphasize that the goal was the early prediction of SAP development, not the differentiation of SAP and NSAP. The text now reads: "There is a lack of effective biomarker combinations capable of predicting SAP from NSAP in the early stages of illness."
Comment 4: A major disadvantage of the study is the absence of samples from healthy donors (control group), which should be explained in the study limitations. By the way, there is no information about ethical approval and informed consent given by the participants in the text.
Response: We have addressed both concerns:
- We have added healthy reference data to Figure 1 (now figure 2), and acknowledged the limitation of not having a proper healthy control group in the Discussion section (Page 10, Lines 290-293).
We thank the reviewer for their attention to ethical standards. We have clarified the ethical status of our study on Page 8, Line 314-318, confirming that this research has received formal approval from the Ethics Committee (protocol code: 2023-IRB-0243-P-01, date of approval: 2023/10/18).
Comment 5: Lines 78–79: What method was used to confirm adenovirus infection?
Response: We have provided a detailed description of the methods used to confirm adenovirus infection in the Methods section (Study Design and Patients,Page 3, Lines 82).
Comment 6: Lines 86–87: How were Th1/Th2 cytokines measured using flow cytometry?
Response: We have provided a detailed description of the equipment used for cytokine detection in the Materials and Methods section, while the specifications of the detection kits have been included in the Appendix B. (Page 11, Lines 334).
Comment 7: Lines 105–111 describe all the obtained results and resemble the Abstract, so they should be removed from the Results section.
Response: We have removed the redundant summary from the Results section and restructured this portion to present the findings in a more appropriate manner (Page 3, Lines 116-127).
Comment 8: Lines 147–150 should be commented in more detail regarding the significance of the obtained AUC values.
Response: We have expanded the discussion on the significance of the AUC values, including detailed explanations of their clinical relevance and evaluation criteria. (Page 5, Lines 168-171).
Comment 9: Figure 1 should be enlarged and resolution should be improved. The figure caption should indicate which panels correspond to different cytokines.
Response: We have regenerated Figure 1 (now figure 2) at high resolution (600 DPI) and enlarged it for better clarity. The figure caption now clearly indicates which panels correspond to specific cytokines and laboratory parameters.
Comment 10: All the figure captions should be made more detailed, including descriptions of the figure elements and color designations.
Response: We have thoroughly revised all figure captions to provide comprehensive descriptions of all elements, including detailed explanations of color codes, symbols, and statistical notations.
Reviewer 4 Report
Comments and Suggestions for Authors
Pathogens-3928935 (article): Clinical and Epidemiological Characteristics of Adenovirus Infection in Children
In this manuscript, Xu et al. take blood samples from Adenovirus positive pediatric patients and perform laboratory assays. They then perform computer analysis to determine an early prediction model of severe adenoviral pneumonia (SAP). Overall the methods section needs to provide more detail as well as some key information added (if available, and if not discuss these limitations in the discussion) for this reviewer to agree on the manuscript being published.
Major comments:
- The methods section needs to be expanded. For the laboratory analysis: What antibodies were used? Kits? Methods for fixing/permeabilizing and staining for flow cytometry, etc. The statistical methods need to be expanded as well: What were in inputs for the programing, etc.? This relates to Figure 2: how were the sensitivities and specificities determined (and this would also be related to the assay).
- Line 77 references a Supplemental Table 1 for diagnosis. There is no supplemental Table 1/how adenovirus was tested for.
- Some key data are left out to determine SAP. For example, which type of Adenovirus did the patient have (if determined), what co-infections were found. While the paper is looking for patient biomarkers, adding this information can be key to determining is a child will develop SAP. If these data are not available, then these limitations need to be discussed.
Minor comment:
Lines 36-37: Adenoviruses cause more than just respiratory diseases, so this sentence needs to be rewritten.
Author Response
Major Comments:
Comment 1: The methods section needs to be expanded. For the laboratory analysis: What antibodies were used? Kits? Methods for fixing/permeabilizing and staining for flow cytometry, etc. The statistical methods need to be expanded as well: What were in inputs for the programing, etc.? This relates to Figure 2: how were the sensitivities and specificities determined (and this would also be related to the assay).
Response: All commercially available kits utilized in this study are documented in the Appendix B, while the R packages employed for statistical analysis are listed in the Materials and Methods section. To facilitate discussion and reproducibility, the statistical code and random forest implementation have been made available in the Supplementary Files.
Comment 2: Line 77 references a Supplemental Table 1 for diagnosis. There is no supplemental Table 1/how adenovirus was tested for.
Response: We thank the reviewer for their careful reading. The mention of "Supplemental Table 1" was a typographical error and has been corrected to Table A throughout the manuscript. The referenced table, which contains the complete diagnostic criteria and adenovirus testing details, is indeed Table A in the appendix.
Comment 3: Some key data are left out to determine SAP. For example, which type of Adenovirus did the patient have (if determined), what co-infections were found. While the paper is looking for patient biomarkers, adding this information can be key to determining is a child will develop SAP. If these data are not available, then these limitations need to be discussed.
Response: We did not perform adenovirus subtyping and only confirmed adenovirus positivity through pharyngeal swabs. Due to the limited number of severe cases collected, patients with co-infections also presented diverse combinations of pathogens. As individual analysis would not yield statistically meaningful groups, we consolidated all cases with three or more pathogens (including adenovirus) into a single category for statistical analysis. We acknowledge this as a current limitation of our study and will continue to enroll more cases in future studies to enhance our research.
Minor Comment:
Comment 4: Lines 36-37: Adenoviruses cause more than just respiratory diseases, so this sentence needs to be rewritten.
Response: We have corrected the relevant statements, revising the generic description to specifically refer to respiratory adenoviral infection (Page 1, Line 35).
Round 2
Reviewer 1 Report
Comments and Suggestions for Authors
The authors have substantially improved the text rendering it suitable for pubblication
Author Response
Comment: The authors have substantially improved the text rendering it suitable for pubblication
Response: Thank you very much for your positive feedback and for acknowledging our efforts in revising the manuscript. We are delighted to hear that you find the revised version suitable for publication.
We greatly appreciate your time and insightful comments, which have been invaluable in improving the quality of our work.
Reviewer 2 Report
Comments and Suggestions for Authors
20 October 2025
Second report manuscript “Clinical and Epidemiological Characteristics of Adenovirus Infection in Children”
Journal Pathogens - MDPI
The revision represents clear progress. The methodological refinements (particularly the implementation of repeated cross-validation, FDR correction, and full code disclosure) have strengthened the credibility of the analysis. The manuscript is now much more transparent and reproducible. Nonetheless, several issues remain that should be corrected before acceptance. These primarily concern (1) harmonization of statistical results and performance metrics across sections,
(2) full consistency between the text and the authors’ FDR statement, (3) correction of minor content and figure errors, and (4) final polishing of language and formatting.
- The Abstract still reports AUC = 0.77; accuracy = 87.8%; precision = 92%, whereas the Results and Table 2 state AUC = 0.699, sensitivity = 91.5%, specificity = 16.7%, and balanced accuracy = 541. In addition, the Discussion mentions a sensitivity of 0.941, not matching Table 2 (0.915). Ensure that all numbers (Abstract, Results, and Discussion) come from the same test-set evaluation. Present sensitivity and specificity alongside accuracy, and explicitly note that performance is moderate due to dataset imbalance. This correction is crucial for scientific consistency and transparency.
- In Section 3.2 the text states that PLT and IFN-γ were significantly different, yet the response to reviewers explains that after FDR correction, only CRP, PA, CPAR, and IL-10 remained significant. Clearly specify in the Results and in Supplementary Table C which p-values are FDR-adjusted and which are raw. The narrative should highlight only variables meeting the FDR < 0.05 threshold if that criterion guided feature selection.
- The caption currently refers to “non-severe/severe acute pancreatitis (NSAP/SAP)”; obviously a leftover from another manuscript. Change to “non-severe/severe adenoviral pneumonia (NSAP/SAP)”.
- In Methods 2.3, the statement “AUC > 0.6 indicates diagnostic utility” is overly strong. Rephrase to “AUC > 6 was considered suggestive of potential diagnostic utility.” Provide 95% CIs for all AUC values.
- The Data Availability section still includes the placeholder “specify the reason for the restriction.” Replace with the actual reason (e.g., patient privacy) or offer anonymized data/code access.
- Funding section still contains “Please ” Supply the finalized funding statement.
- Remove residual template text (“Type of the Paper (Article)” and editorial headers/footers).
- Given the severe class imbalance (SAP:NSAP = 54:374), emphasize balanced accuracy, F1-score, and, ideally, Matthews correlation coefficient (MCC). Adding bootstrap confidence intervals for these metrics in Table 2 would further strengthen the results.
- Several biomarkers (e.g., IL-6, TNF-α, IL-10) show extreme variance (see Supplementary Table C). Report both mean ± SD and median [IQR], or apply log transformation and use non-parametric testing. Clarify your outlier-handling policy.
- The “within 24 h of admission” inclusion criterion does not necessarily control for time since symptom onset. Add this limitation explicitly to the Discussion.
- In Figure 2, add the sample size (n) per group on each panel; ensure consistent units on y- axes (ratios may remain unitless). In Figure 3, add 95% CIs for AUCs and indicate the number of cases used to compute each curve.
- The manuscript has improved markedly after professional editing, yet a few issues persist:
- Abbreviations: Correct “C-reactive Protein c” and “Interferon-Γ” to “C-reactive protein (CRP)” and “Interferon-γ (IFN-γ).”
- Grammar: “Comparing with non-severe cases” → “Compared with non-severe ”
- Flow: Remove repeated sentence structures (“This study identifies… Indicators are valuable… The model demonstrated…”). Focus instead on the clinical implications and limitations already well acknowledged (single center, no genotype or viral load data).
The remaining issues are limited to consistency, clarity, and presentation. Once the authors align all statistical metrics, correct the mislabeled figure, specify FDR adjustments, and finalize minor formatting and language edits, the manuscript will be suitable for acceptance in Pathogens.
Comments on the Quality of English Language
The English quality of the revised manuscript is satisfactory but not yet fully publication-ready. The authors have clearly improved grammar, sentence structure, and overall readability since the previous version, and the paper is now easy to follow in most parts. However, the text still contains several stylistic and grammatical issues that require careful polishing by a professional editor before final acceptance.
Verb tenses are used inconsistently: past and present forms alternate in the Methods and Results sections, which affects clarity. Articles and prepositions are occasionally missing or misused (for example, “in presence of” should be “in the presence of”). Many sentences remain long and complex, connecting multiple ideas with commas, which results in run-on structures and slightly awkward flow. Simplifying these into shorter, more direct statements would greatly improve readability.
The vocabulary is scientifically accurate but sometimes not idiomatic. Expressions such as “showed obvious increase” or “proved the importance” should be replaced with more natural academic phrasing like “showed a marked increase” or “suggested the importance.” Similarly, certain phrases are repeated verbatim across sections and could be reworded for stylistic variety. The tone is mostly appropriate, but at times overly assertive given the study’s limitations. Words like “demonstrated” or “proved” should be replaced with “indicated” or “suggested” to maintain a balanced and objective style. Figure legends and table captions are understandable but would benefit from grammatical tightening and consistent abbreviation use.
Overall, the manuscript is written at an upper-intermediate level of English. The content is clear and coherent, but the paper still needs professional language editing to refine syntax, punctuation, and flow. After these adjustments, the text will fully meet the linguistic standards of Pathogens. The manuscript is comprehensible and scientifically sound, but requires final professional copy-editing to achieve publication-level fluency. A professional copy-edit by a native or near-native English editor (ideally one familiar with biomedical writing) is strongly recommended.
Author Response
- Comment: The Abstract reports AUC=0.77, Accuracy=87.8%, Precision=92%, whereas the Results section and Table 2 state AUC=0.699, Sensitivity=91.5%, Specificity=16.7%, and Balanced Accuracy=54.1%. Furthermore, the Discussion mentions a sensitivity of 0.941, which does not match Table 2 (0.915). Ensure all numbers (Abstract, Results, Discussion) originate from the same test-set evaluation. Present sensitivity and specificity alongside accuracy, and explicitly note that the performance is moderate due to dataset imbalance. This correction is crucial for scientific consistency and transparency.
Response: Thank you for this correction. We have unified all the relevant data throughout the manuscript (see Page 1, Lines 24; Page 6, Line 186; Page 9, Line 270) and have provided a more accurate evaluation of the model's performance (Page 9, Lines 271-274).
- Comment: Section 3.2 states that PLT and IFN-γ were significantly different, yet the response to reviewers explains that after FDR correction, only CRP, PA, CPAR, and IL-10 remained significant. Clearly specify in the Results and in Supplementary Table C which p-values are FDR-adjusted and which are raw. The narrative should highlight only variables meeting the FDR < 0.05 threshold if that criterion guided feature selection.
Response: Your suggestion has significantly improved the quality of our manuscript. We have linked the Wilcoxon test results with the random forest evaluation in our description (Page 7, Lines 191-196), discussed the reasons for the discrepancies and the final feature selection principle (Page 9, Lines 278-287), and added the FDR-adjusted p-values to Supplementary Table C for reference (Page 12, Table C).
- Comment: The caption currently refers to "non-severe/severe acute pancreatitis (NSAP/SAP)"; obviously a leftover from another manuscript. Change to "non-severe/severe adenoviral pneumonia (NSAP/SAP)".
Response: We thank the reviewer for their careful review. This error was indeed an oversight and has now been corrected (Page 7, Lines 203-204).
- Comment: In Methods 2.3, the statement "AUC > 0.6 indicates diagnostic utility" is overly strong. Rephrase to "AUC > 0.6 was considered suggestive of potential diagnostic utility." Provide 95% CIs for all AUC values.
Response: We agree that the original statement was too assertive. We have rephrased it to a more conservative interpretation (Page 3, Lines 104-105) and have provided 95% confidence intervals for all AUC values.
- Comment: The Data Availability section still includes the placeholder "specify the reason for the restriction." Replace with the actual reason (e.g., patient privacy) or offer anonymized data/code access.
Response: We thank the reviewer for this guidance. Following discussion, we have revised this section to use a more standard and compliant formulation (Page 11, Line 345).
- Comment: The Funding section still contains "Please". Supply the finalized funding statement.
Response: We apologize for this oversight. The funding section has now been correctly completed (Page 10, Lines 334-338).
- Comment: Remove residual template text ("Type of the Paper (Article)" and editorial headers/footers).
Response: All template text, headers, and footers have been removed as requested.
- Comment: Given the severe class imbalance (SAP:NSAP = 54:374), emphasize balanced accuracy, F1-score, and, ideally, Matthews correlation coefficient (MCC). Adding bootstrap confidence intervals for these metrics in Table 2 would further strengthen the results.
Response: The requested information has been added to Table 2 (Page 7, Line 206).
- Comment: Several biomarkers (e.g., IL-6, TNF-α, IL-10) show extreme variance (see Supplementary Table C). Report both mean ± SD and median [IQR], or apply log transformation and use non-parametric testing. Clarify your outlier-handling policy.
Response: We thank the reviewer for this guidance. We have added the median [IQR] to Table C.
- Comment: The "within 24 h of admission" inclusion criterion does not necessarily control for time since symptom onset. Add this limitation explicitly to the Discussion.
Response: This guidance is crucial for the rigor of our manuscript and for informing future model refinement. We have explicitly added this limitation to the Discussion section (Page 10, Lines 301-306), and our future patient data collection methods will be adjusted accordingly. Thank you.
- Comment: In Figure 2, add the sample size (n) per group on each panel; ensure consistent units on y-axes (ratios may remain unitless). In Figure 3, add 95% CIs for AUCs and indicate the number of cases used to compute each curve.
Response: We thank the reviewer for this guidance. We have improved Figures 2 and 3 accordingly (Page 6). Regarding "ensure consistent units on y-axes," our understanding is that different indicators have inherently different measurement methods (e.g., cell count, substance concentration) and units used in clinical practice. We have ensured clear and consistent unit labeling within respective figure panels, with ratio indicators being unitless.
- Comment: The manuscript has improved markedly after professional editing, yet a few issues persist:
- Abbreviations: Correct "C-reactive Protein c" and "Interferon-Γ" to "C-reactive protein (CRP)" and "Interferon-γ (IFN-γ)."
Response: We thank the reviewer for their meticulous proofreading. These two errors have been corrected (page 11).
- Grammar: "Comparing with non-severe cases" → "Compared with non-severe cases."
Response: This expression has been revised. (Page 10, Line 323)
- Flow: Remove repeated sentence structures ("This study identifies… Indicators are valuable… The model demonstrated…"). Focus instead on the clinical implications and limitations already well acknowledged (single center, no genotype or viral load data).
Response: Regarding the flow, we appreciate the guidance. The relevant phrasing has been changed. We have dedicated a specific paragraph in the Discussion section to emphasize the study's limitations and indicated that subsequent research will focus on improving these aspects (Page 10, Lines 298-314).
Reviewer 4 Report
Comments and Suggestions for Authors
All prior comments are addressed
Author Response
Dear Reviewer,
We sincerely thank you for your overall positive assessment of our revised manuscript and for the valuable suggestions to further enhance the quality of our work. We have carefully addressed each of your specific comments point-by-point in the latest version of the manuscript. The detailed responses are provided below:
- Comment: Are the methods adequately described? (Must be improved).
Response: We have added Appendix Table B to detail the reagents and equipment used for biomarker detection. The supplementary material now includes all R code utilized in this study. - Comment: Are the conclusions supported by the results? (Must be improved).
Response: We have moderated claims regarding the model's capabilities in the conclusion to ensure they align more accurately with the actual results. - Comment: Is the research design appropriate? (Can be improved).
Response: To address issues such as the small sample size and class imbalance, we adapted the random forest design by incorporating FDR correction and cross-validation. We also acknowledge the design limitations in the Discussion section and commit to avoiding these issues in future studies through improved methodologies. - Comment: Are the results clearly presented? (Can be improved).
Response:We have enhanced the clarity and comprehensiveness of our results by adding the sample size for each group in Figure 2, confidence intervals and sample sizes for ROC curves in Figures 3 and 4, and additional model performance parameters in Table 2. Appendix Table C now includes Median (IQR) and Adjusted P-Value.
Sincerely,
Huifen Xu
On behalf of all co-authors